# Narrative Review of the Control and Prevention of Knowlesi Malaria

**DOI:** 10.3390/tropicalmed7080178

**Published:** 2022-08-11

**Authors:** Ahmad Hazim Mohammad, Nurul Athirah Naserrudin, Syed Sharizman Syed Abdul Rahim, Jenarun Jelip, Azman Atil, Mohd Fazeli Sazali, Adora J. Muyou, Priya Dharishini Kunasagran, Nornazirah Ahmad Kamarudin, Zahir Izuan Azhar, Rahmat Dapari, Mohammad Saffree Jeffree, Mohd Rohaizat Hassan

**Affiliations:** 1Department of Public Health Medicine, Faculty of Medicine and Health Sciences, Universiti Malaysia Sabah, Kota Kinabalu 88400, Malaysia; 2Department of Community Health, Faculty of Medicine, Universiti Kebangsaan Malaysia, Kuala Lumpur 56000, Malaysia; 3Sabah State Health Department, Ministry of Health Malaysia, Kota Kinabalu 88590, Malaysia; 4Disease Control Division, Ministry of Health Malaysia, Putrajaya 62590, Malaysia; 5Department of Public Health Medicine, Faculty of Medicine, Universiti Teknologi MARA (UiTM), Sungai Buloh 47000, Malaysia; 6Department of Community Health, Faculty of Medicine and Health Sciences, Universiti Putra Malaysia, Serdang 43400, Malaysia; 7Borneo Medical and Health Research Centre, Faculty of Medicine and Health Sciences, Universiti Malaysia Sabah, Kota Kinabalu 88400, Malaysia

**Keywords:** malaria, *Plasmodium knowlesi*, prevention, control, review

## Abstract

Despite the reduction in the number of cases of human malaria throughout the world, the incidence rate of knowlesi malaria is continuing to rise, especially in Southeast Asia. The conventional strategies for the prevention and control of human malaria can provide some protection against knowlesi malaria. Despite the numerous studies on the risk factors and the innovative methods that may be used to prevent and control the vectors of *Plasmodium knowlesi,* the incidence rate remains high. An integrated approach that includes environmental intervention should be adopted in order to ensure the successful control of zoonotic malaria. A combination of personal-level protection, vector control and environmental control may mitigate the risk of *Plasmodium knowlesi* transmission from macaques to humans and, ultimately, reduce the incidence rate of knowlesi malaria.

## 1. Introduction

*Plasmodium knowlesi* has become one of the leading causes of malaria in humans, especially throughout Southeast Asia [1]. In 1931, the parasite was first isolated from a long-tailed macaque (*Macaca fascicularis*) by an Italian malariologist [2]. Later, *Plasmodium knowlesi* was found to cause naturally acquired malaria in pig-tailed macaques (*Macaca nemestrina*) and the mitred leaf monkey (*Presbytis melalophos*) [3]. In 1932, a group of researchers found that the parasite is capable of causing infection in humans [4]. In 1965, the first naturally acquired human case was reported in Pahang, Malaysia [5], and the second natural infection of *Plasmodium knowlesi* was reported in Johor, Malaysia, in 1971 [4].

Given the fact that there is no evidence of human-to-human transmission, the human is considered to be a dead-end host for *Plasmodium knowlesi* [6,7,8]. Though there is the possibility of peri-domestic transmissions and among household members, these cannot be directly attributed to human-to-human transmission [9]. To prove that human-to-human transmission is possible, researchers may need to feed mosquitoes with infected human blood through a membrane feeder and then determine if sporozoites can develop [10]. The majority of human *Plasmodium knowlesi* cases were exposed to female *Anopheles* mosquitos harbouring the *Plasmodium knowlesi* in the forest or at the edges of the forest and plantations [11]. That is the reason why most of the knowlesi malaria cases have been reported among people with forest-related jobs or those who were doing activities that are related to the forest [12]. The majority of the cases were reported among males of ages from 20 to 50 and the occupations which are known to pose the highest risk of infection were in the agricultural and plantation line [8,13]. It was found that the incidence rate of knowlesi malaria was higher among people with low-level education as compared to people with a higher level of education, as the individuals belonging to the latter group were much more aware of the preventive measures against knowlesi malaria infection [14].

Globally, there was a total of 241 million malaria cases reported in 2020, which is an increase from 2019 which saw 217 million cases reported. Over 90% of the global malaria cases were reported in the African Region. In the Western Pacific region, cases of malaria have been reducing in their frequency for the past 10 years; however, in some countries, there was a rise in the incidence rate of zoonotic malaria [15]. Malaysia is the country with the highest number of reported *Plasmodium knowlesi* cases compared to other countries in Southeast Asia. From 2010 to 2018, there have been over 18,000 cases reported in Malaysia [16]. The sharp rise in the incidence of *Plasmodium knowlesi* in Malaysia, especially in Sabah and Sarawak, was attributed to the improvement of molecular diagnostics and increasing awareness among the population. Despite the success of the National Malaria Elimination Programme, with zero indigenous human malaria cases since 2018, the dynamic increase of human *Plasmodium knowlesi* is a public health threat [17]. *Plasmodium knowlesi* cases were also reported in children and women following the increase in peri-domestic transmission [18].

The number of cases might be underreported as there is also evidence that has suggested that persons who were infected by *Plasmodium knowlesi* can be asymptomatic, as was reported by a case–control study that was undertaken in Sabah [9]. A study that was undertaken in Cambodia also supports the notion that zoonotic malaria cases could be asymptomatic as there were about 21 cases that were detected among people who did not have a history of fever or any other complaints. Of these, 8 out of the 21 cases were confirmed as having been caused by knowlesi malaria [19]. The presence of asymptomatic cases could indicate that there is widespread transmission among the communities that are in high-risk areas. The current strategies for prevention and control are more often directed towards domestic and peri-domestic transmissions; however, the majority of the affected individuals acquired the infection on a farm or in the jungle [17] so we must enhance the self-preventive actions among the community, such as wearing long sleeves and long pants, using mosquito repellents and even avoiding vector active time in order to protect from knowlesi malaria infection.

Apart from Malaysia, other countries in Southeast Asia, including Thailand, Myanmar, Vietnam, Indonesia and the Philippines, had reported significant numbers of knowlesi malaria cases throughout the years [1,20,21,22,23,24]. The method that is used for diagnosing knowlesi malaria in the field, especially in resource-limited areas, is through the use of a blood film for malaria parasites (BFMP) [25]. However, misdiagnosis has always occurred due to the similarities in the morphology of *Plasmodium knowlesi* to *Plasmodium malariae* and *Plasmodium falciparum* when they are viewed under a microscope [26,27]. Hence, molecular methods, which include polymerase chain reaction (PCR), are the gold standard for diagnosing knowlesi malaria due to their higher sensitivity and specificity while rapid diagnostic tests like LAMP may offer a rapid, simple and reliable test for the diagnosis of malaria in areas where malaria is prevalent [28]. Another promising diagnostic method that can be implemented in rural areas is rotating-crystal magneto-optical detection (RMOD). A study of this method that was conducted in Papua New Guinea showed comparable sensitivity and specificity to the current standard diagnostic methods; however, it was never tested to detect *Plasmodium knowlesi*. Theoretically, it may detect *Plasmodium knowlesi* as RMOD principally targets hemozoin, but further studies on this notion are needed [29]. The lack of rapid, affordable and accurate diagnostic tests represents the primary hurdle that is affecting malaria surveillance in resource- and expertise-limited areas [28].

Early detection and prompt treatment of knowlesi malaria are essential to prevent severe complications, such as acute kidney failure, respiratory distress [24,25,26] and intravascular haemolysis [30,31,32,33]. There have also been a few reports of rare complications of knowlesi malaria, such as splenic rupture [34] and spontaneous pulmonary haemorrhage [35]. Despite reports of transfusion-transmitted malaria (TTM) being rare in the case of *Plasmodium knowlesi*, it has been reported to result in severe infections [36]. The acquisition of malaria in the form of TTM has led to the suggestion for a more thorough screening process to be implemented prior to the provision of blood transfusion services [36]. Knowlesi malaria among women is rare; however, there are reported cases wherein it has caused anemia as well as low-birth-weight infants [37]. It is important to screen for and detect early among pregnant women, particularly those who live in a high-risk area. We can also look into the use of intermittent preventive treatment (IPT) among pregnant women as this treatment has demonstrated a reduction in the prevalence of low-birth weight infants [38].

*Plasmodium knowlesi* is capable of causing severe infections in humans, infections which are similar to those that are caused by *Plasmodium falciparum* due to its short 24 h asexual cycle which leads to hyperparasitemia [39]. Knowlesi malaria can be fatal if it is detected and treated late. There have been fatalities reported in Malaysia [40] and Thailand [41] that were due to late detection and misdiagnosis. The treatment of knowlesi malaria is very similar to that of *Plasmodium falciparum* and *Plasmodium malariae* [42]. Knowlesi malaria can be effectively treated with the current antimalarial drugs that are available, which include chloroquine [43,44,45] and artemisinin combination therapy (ACT). Currently, there is still no evidence of drug-resistant knowlesi parasites [42,45,46]. For severe infections, the prior administration of intravenous artesunate is effective to reduce the risk of severe and fatal complications [47,48].

When considering the severe and fatal complications of knowlesi malaria, it is important to emphasize the adoption of preventive measures in order to reduce the risk of infection among people in high-risk areas, especially those who live in rural regions, in order to reduce the possibility of mortality. In addition, the lack of proper diagnostic tools in rural areas makes it difficult to diagnose knowlesi malaria, hence misdiagnosis or late detection may occur, which will later result in late treatment. Furthermore, people in rural areas might not be aware that knowlesi malaria has been detected in their region, as was shown by a study that was undertaken among aboriginals and rural people in Pahang, Malaysia that found that their knowledge and practice of preventive measures are inadequate; however, this particular study is mainly about human malaria [49]. Hence, this may contribute to late healthcare-seeking among rural communities. A health promotion program, such as a door-to-door information dissemination campaign, would improve the knowledge of prevention and early healthcare-seeking among rural communities. Thus, the main objective of this review is to appraise the control and preventive measures that are available that might be able to reduce the transmission of knowlesi malaria among the effected communities, which in turn will reduce the risk of severe or mortality cases.

## 2. Prevention and Control of Knowlesi Malaria

The current tools that are used for the prevention and control of knowlesi malaria are mostly adapted from the control of human malaria. However, the transmission dynamics of zoonotic *Plasmodium knowlesi* require an integrated approach, such as that which is detailed in the “One Health” concept. The approach details the management of infectious disease, while using a collaborative, multisectoral and transdisciplinary approach with the goal of achieving optimal health outcomes for people, animals and their shared environment [1]. Control and prevention of knowlesi malaria require multi-sectoral collaboration, not only with the government and stakeholders, but also with the wildlife department. The involvement of macaques as the natural host of *Plasmodium knowlesi* poses a challenge to the elimination and subsequent eradication of knowlesi malaria and requires a more complex intervention measure as compared to intervention against human malaria [50].

### 2.1. Indoor Residual Spraying (IRS) and Outdoor Residual Spraying (ORS)

Currently, the prevention and control of knowlesi malaria is mainly focused on vector control methods, which include insecticide spraying for both indoors and outdoors as well as perimeter spraying (which is the spraying of insecticide in a five-meter radius area surrounding the house) [51]. Indoor residual spraying (IRS) involves coating the walls and other surfaces inside the house with an insecticide in order to purposely kill the adult *Anopheles* mosquitoes that rest on the walls after they bite people. In order to ensure the effectiveness of IRS, the insecticide must cover at least 80% of the walls inside the house [52]. The insecticide will last for around three to six months, depending on the type of surface of the wall of the house. Rural or traditional houses that are built with wood with some gaps may need to be sprayed regularly every three months [53].

IRS is the primary method that was used during the Global Malaria Eradication Campaign from 1955 to 1969. At the time, DDT was used as the insecticide to kill mosquitoes; however, due to the environmental impact of DDT, it was replaced by a newer type of more expensive insecticide. Subsequently, IRS was abandoned due to the cost; however, later, in Africa, IRS was successfully implemented and resulted in the revival of the method [52]. In Malaysia, deltamethrin is used as an insecticide for indoor residual spraying which primarily targets the mosquitoes that rest indoors and outdoors. Apart from deltamethrin, the WHO also recommends alpha-cypermethrin, permethrin and cyfluthrin. These insecticides are more effective in controlling the mosquitoes that rest and bite indoors but they are less effective against outdoor biters [54]. Given that the vectors of knowlesi malaria are outdoor biters, the efficacy of IRS as vector control for *Plasmodium knowlesi* is doubtful. Novel strategies are essential for the control of zoonotic malaria [11]. There have been reports of IRS resistance, such as that which occurred in Burkina Faso; however, the resistance was only towards DDT while organophosphates remained effective [55]. Currently, there are no reports of insecticide resistance among knowlesi vectors.

The bionomics of the *Plasmodium knowlesi* vectors are currently not fully understood. However, the current knowledge is that the vectors of *Plasmodium knowlesi* belong to the *Leucosphyrus* group of *Anopheles* mosquitoes. They are generally described as forest-dwelling mosquitoes [56]. Among the implicated *Anopheles* species are *Anopheles cracens, Anopheles maculatus* and *Anopheles introlatus* in Peninsular Malaysia [57], *Anopheles latens* and *Anopheles donaldi* in Sarawak [58,59] and *Anopheles balabacensis* in Sabah [60,61,62], which also can be found in Sarawak [58]. Elsewhere, *Anopheles dirus* is the vector in Vietnam [63]. *Anopheles dirus* can be both exophagic and endophagic and the biting time usually starts before 21.00 h [63]. *Anopheles latens* and *Anopheles cracens* are more commonly found in forests and farms than in villages; while *Anopheles balabacensis* are found in all areas, i.e., farms, forests and villages [56]. *Anopheles latens* can be an indoor or outdoor biter with active biting time as early as 18.00 h up until dawn [59] while *Anopheles balabacensis* are specifically outdoor biters with active biting time also starting at dusk up until dawn [57]. *Anopheles cracens*, which is the main vector for *Plasmodium knowlesi* in West Malaysia, are predominantly exophagic and the mosquitoes are mostly active before 21.00 h [57]. For this reason, researchers had conducted experimental outdoor residual spraying (ORS) and perimeter spraying using a new formulation of deltamethrin in Sarawak and Sabah [64,65].

In these studies, the outdoor residual spraying was done by spraying an insecticide onto the exterior wall surfaces of building or houses while the perimeter spraying was conducted on the trunks of the trees which were located around a five-meter radius from the house [51]. Based on the results of the study in Sabah, it has been suggested that the newly formulated deltamethrin could effectively maintain an 80% mortality rate of *Anopheles balabacensis* for at least six months [64]. A similar study was conducted in the rural areas of Sarawak and it also reported that this method was able to reduce the population of the main vectors for malaria in Sarawak, including *Anopheles donaldi, Anopheles latens* and *Anopheles introlatus* as well as *Anopheles balabacensis* [65].

Even though the trials showed some efficacy in reducing the population of the mosquitoes, they have not effectively been shown to reduce the incidence rate of knowlesi malaria. Furthermore, it was found that most of the cases were acquired somewhere away from their home, such as when the effected individuals were sleeping outdoors [14]. The *Anopheles balabacensis* mosquitoes can cover a range as large as 500 m, which is a very vast area, and it was found that these mosquitoes fly in many areas near the jungle such as the forest edge and oil palm plantations [66].

### 2.2. Insecticide-Treated Nets (ITNs)

The use of insecticide-treated nets (ITNs) and long-lasting insecticide nets (LLINs) has been proven to protect susceptible groups from getting human malaria per reports from a few studies that were done in Africa [67] and Asia [68]. The insecticide can last for 3 years and the nets are conventionally treated. The insecticide remains active for 12 months [69]. Bed net use is one of the main preventive methods that can be used to protect people from the bites of *Anopheles* mosquitoes [70]. Assuming that simian malaria shares the same vectors as human malaria, the use of bed nets would offer similar protection; however, ITNs were found to be less protective mainly due to the exophagic behaviour of the mosquitoes [14]. There has been concern about resistance to ITN among the malaria vectors as the reduction of global malaria cases was stalling, especially in the African region; however, based on the review by Lindsay et al., there has been little evidence suggesting resistance towards ITN [71]. We may need to assess the compliance, bioefficacy and durability of the nets among the communities in order to confirm the concern.

### 2.3. Larval Source Management (LSM)

Larval source management (LSM), which is a less popular method of control, is the management of the habitats that are required for mosquito breeding, which include water bodies such as ponds, pools and rivers. In order to deal with the outdoor biters, such as *Anopheles balabacensis,* LSM could provide the benefit of reducing the population of the mosquitoes. There are a few methods of LSM that are used, such as habitat modification, habitat manipulation, biological control and larviciding [69]. The WHO recommends LSM where the methods are feasible and effective in disrupting the stage of development from larva to adult mosquitoes as this may reduce the risk of malaria parasite transmission [70].

Habitat modification is a permanent adjustment to the natural environment of the mosquitoes, which includes making a change to the land and water such as filling, drainage and land reclamation. By filling water bodies such as ponds, pools and puddles, as well as draining the water out of them, it is possible to reduce the population of mosquitoes by removing the potential breeding sites for the mosquitoes’ larvae [72]. Habitat modification can easily be conducted in a rural area, such as, for example, those that are found in India wherein the incidence rate of malaria in the area had significantly reduced compared to before the intervention [73]. While it is potentially beneficial, the effectiveness of this method in areas with a high incidence of knowlesi malaria is still unclear. Habitat modification is possible to be implemented in the peri-domestic regions and near farms but implementing habitat modification in the jungle would not be feasible. However, *anopheline* mosquitoes are more prevalent in the jungle [74]; therefore, this method would be less effective in reducing the vector population.

Habitat manipulation, on the other hand, is the temporary alteration of the potential breeding sites of the vectors. This method may need to be done repeatedly and it is suitable for use in areas with limited resources. The method includes flushing and draining [69]. Habitat manipulation, when combined with regular larviciding and habitat modification, has successfully reduced the incidence of malaria in Greece, India, Philippines and Tanzania [73]. A study that was undertaken in Ethiopia that manipulated a water level by drawing down the water in a few dams to the nearby plantation reported that the incidence rate of malaria in that area was markedly reduced; however, this method of manipulation needs to be sustained in order to ensure the successful control of the vectors [75].

Biological control is the method wherein we introduce predators of mosquito larvae, such as predatory fish. Biological control needs to be conducted properly and must be supervised by experts in the field [69]. Although introducing larvivorous fish into the natural environment of the mosquitoes may lead to a reduction in the larvae counts, numerous studies that were undertaken in various countries in Asia and Africa, including India, Sri Lanka, Korea, Sudan, Ethiopia and Kenya, found that the method was not effective in reducing the population of *Anopheles* mosquitoes [76]. This method requires a large resource commitment and its efficacy for the vectors of knowlesi malaria remains unclear.

The most common larval source management method that is used in many malaria-endemic countries is larviciding by applying a biological or chemical insecticide to water bodies [70]. Larviciding is suitable for the small number of permanent and obviously visible habitats. Microbial larvicide is effective in controlling the mosquito population and it is much safer than chemical larvicide; however, this method is more expensive [69]. Studies that were conducted in African countries using insect growth regulators reported a reduction in the incidence rate of malaria and the prevalence of the parasites throughout the study period [77]. Larviciding needs to be done regularly in order to ensure the effectiveness of the method in keeping the mosquitoes’ population low.

### 2.4. Housing Improvement

Housing is one of the main environmental factors that play a role in the transmission of malaria parasites. Wooden rural-type houses may have gaps in the walls and roofs creating an opening for the mosquitoes to enter the house, thus resulting in a higher risk of getting malaria. Modern houses have proven to be more protective than the traditional rural types of houses and, in some of the settings, the protection can be considered as equivalent to the protection that is provided by ITNs [69]. By closing the gaps between doors, walls and roofs, the risk of contact with *Anopheles* mosquitoes can be reduced. Although the main vectors for knowlesi malaria are mostly exophagic, the mosquitoes were found to go inside houses as well [17]. House improvement can be applied if there are affordable options for the people [78].

### 2.5. Personal-Level Protection

Personal-level protection is mainly intended to prevent people from sustaining mosquito bites, thus reducing their risk of getting a malaria infection. There are three main interventions that are included as part of personal protection, these are topical repellent, insecticide-treated clothing and spatial repellent [79]. Personal-level protection, such as spatial repellent, has been used for centuries for mosquito bite prevention and the WHO recommends that travellers use these methods of protection when traveling to malaria-endemic areas [80]. The mode of action for each intervention varies; however, they are generally effective in preventing mosquito bites, especially when used during outdoor activities.

The primary function of topical repellents is to interfere with the olfactory reception of mosquitoes, which results in them having difficulties locating and biting humans. Topical repellents are available in various formulations such as gels and lotions. They contain a wide range of active ingredients which include N, N-diethyl-m-toluamide (DEET), icaridin and picaridin [79]. A study that was undertaken in Myanmar found that topical repellents offer some protection in reducing the risk of malaria infection; however, compliance was required among the users in order to ensure the effectiveness [81]. A study that was undertaken in Cambodia showed that topical repellent use among the participants was suboptimum, hence it did not contribute towards a reduction in the incidence rate of malaria [82]. Some of the reasons that affected the compliance were related to perceptions that repellents are toxic and not safe, especially for children. Furthermore, many of the study’s participants reported skin-related conditions, such as rash and dry skin [83].

Insecticide-treated clothing (ITC) has been used by military personnel and travellers for years as a means of personal protection against mosquito bites. The most common insecticide that is used for clothing is permethrin, but there are also several other, less common insecticides that can be used, such as bifenthrin, deltamethrin and cyfluthrin [84]. The WHO approves the usage of permethrin as it is much safer compared to other insecticides because it has low dermal absorption, minimal odour, minimal irritation and low mammalian toxicity [85]. The mosquitoes supposedly make oriented movement away from the treated people after coming into contact with ITC [79]. Based on various studies that were undertaken in order to investigate its efficacy and effectiveness, it was reported that ITC may protect against mosquito bites [79]. Studies on ITC have focused more on protection against human malaria parasites; hence, it is still unclear whether ITC will offer similar protection against zoonotic malaria. There was no issue with compliance as there was limited clothing options for military personnel and refugees [79].

Spatial repellent may be able to prevent mosquitoes entering a space that is occupied by humans and thus reduce the contact between humans and vectors. There are more than 10 active ingredients that have been approved by the WHO for use as spatial repellent, including pyrethroid [86]. Pyrethroid acts by affecting the nervous system of the mosquitoes [79]. Based on studies that were undertaken in Indonesia, spatial repellent did not have a significant impact in reducing the population of mosquitoes [87]. Spatial repellent also can only target the indoor biters, while the vectors of *Plasmodium knowlesi* tend to be outdoor biters [56].

### 2.6. Novel Methods for Prevention and Control

The odour-based mosquito trap is one of few novel approaches that can be used to reduce the mosquito population. Based on a study that was conducted in Australia, a trap which is fan-powered and baited with CO_2_ was able to attract and trap high numbers of mosquitoes [88]. This method could be a promising addition to the existing vector control modalities. Other than the trapping method, an odour attractant that is laced with toxic sugar that can kill both male and female mosquitoes could be utilized in order to reduce the mosquito population [89]. A large-scale field trial in Mali reported that the population of both female and male mosquitoes was significantly reduced after using this method [90].

### 2.7. Community Participation

Anthropogenic activities like plantation, farming and logging activities and deforestation have resulted in ecological changes that could change the behaviour of macaques and the knowlesi malaria vectors. The ecological changes may result in the disruption of the natural food sources for macaques and force them to migrate to places with a greater food supply that are in close proximity to human residential areas. This will result in close encounters between the macaques and humans which will lead to an increased risk of parasite transmission [74]. The ideal solution is to regulate deforestation activities, which requires strong political will from all of the agencies that are involved.

Several countries, such as India, Myanmar, Nigeria and Malaysia, have adopted a community participation approach including the recruitment of volunteers from the village and community members for malaria control programs [91,92,93,94,95]. The volunteers are directly involved in the collection of blood films for malarial parasite (BFMP) and performing RDT among villagers who exhibit symptoms of malaria infection and they are also responsible to convey this information to the local healthcare system [93,96]. Community volunteers also play a role in the distribution of LLINs, especially in rural malaria-effected areas [97]. They are given training in order to perform their tasks and they are entrusted with the responsibility to protect their community. Some of the other tasks that are entrusted to them are to convey information on malaria and the identification of people who are at risk, risky activities and personal protection against malaria infection. The roles that are played by the volunteers ensure access to early detection and the ability for individuals to acquire information on the prevention of knowlesi malaria infection, even in the most remote villages. In essence, positive deviance that is instilled through community volunteers may change community behaviour and indirectly reduce knowlesi malaria transmission [98].

Given the complex transmission dynamic of knowlesi malaria, interim strategies are needed in order to control knowlesi malaria transmission while waiting for a more effective strategy to be discovered. Apart from enhancing and optimising the known vector control strategies, case detection and management, community participation in *Plasmodium knowlesi* control should be optimized. Horizontal approaches involving community engagement may contribute to a more sustainable malaria control programme [99] and this will hold great benefit during pandemic conditions, such as that of COVID-19 wherein the healthcare staff could not go to the malaria-effected areas for regular spraying and other preventive measures due to the lockdown. This inability for healthcare staff to enact preventative measures can cause an increase in the incidence rate of malaria, including that of knowlesi malaria [100].

## 3. Conclusions

The high incidence rate of knowlesi malaria might pose significant challenges to the progress of malaria elimination, especially in Southeast Asia. Conventional preventive and control strategies have been shown to provide some degree of protection among the susceptible population. However, given the more complex transmission dynamic of knowlesi malaria, an integrated control approach involving multisectoral collaboration at the national and global levels is required. In other words, we need to adopt a One Health approach which may include respective national health authorities in order to prevent transmission in the respective countries as well as global transmission, especially among travellers. There is a need to explore and adopt new innovative methods of prevention and control in order to reduce the incidence rate of knowlesi malaria. The most important aspects to consider in selecting new preventive methods are their effectiveness, acceptability, affordability and sustainability in order to ensure successful control. We can never exclude environmental factors, such as forest conservation, as parts of the strategy for reducing the burden of zoonotic malaria. Lastly, we need to empower our community to contribute to the efforts in order to prevent and control knowlesi malaria.

## Data Availability

Data sharing not applicable.

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
