# Peer review of "Narrative Review of the Control and Prevention of Knowlesi Malaria"

_tropicalmed, 2022, doi:10.3390/tropicalmed7080178_

Round 1

Reviewer 1 Report

It is a very interesting review about control and prevention of zoonotic malaria by P. knowlesi, a Plasmodium of monkeys of Southeast Asia. The manuscript, in general, is well written and just needs a few modifications that will be more detailed below. The authors informed that this is a “narrative review”, so the rules about systematic reviews and meta-analysis do not apply for this manuscript.

Major comments:

1)                  Although malaria by P. knowlesi can produce serious illness and death, this is not the only argument for thinking about measures to control and prevent this parasite. The planet is committed to the elimination of malaria and in this sense, thinking about prevention and reduction of P. knowlesi transmission is essential. The first argument (to reduce the possibility of mortality cases) was addressed by authors, but the second (and maybe more important in this topic) is discussed too superficially. It is necessary to separate the two approaches: 1) Control and prevention measures for transmission (the manuscript presents fundamentally these questions) and 2) Control and prevention measures for avoiding severe cases and deaths for P. knowlesi infections.  

The manuscript will benefit if the authors clearly show these two approaches by presenting the existing evidence on each topic, like this:

1.1.                   Evidence about transmission at the community level (as shown) but adding information about the presence of asymptomatic cases and mild malaria. Data on the percentage of asymptomatic individuals and patients with mild malaria by P. knowlesi, for example. The feeling when reading the article is that most cases of P. knowlesi malaria correspond to serious illness and death. More information about risk factors for infection is desirable. What are the main measures for the control and prevention of this malaria that occurs in the community context? if the strategies are successful, the incidence of the disease could decrease. The authors devoted much part of their review to this very important topic, but they were not successful in showing what part of the P. knowlesi malaria burden would be met with these strategies.

1.2.            On the other hand, they should show evidence for control and prevention of severe illness cases and death by P. knowlesi. What is known about the serious disease burden and death of patients infected with this parasite? What are the main measures for the control and prevention of severe malaria and death by P. knowlesi? How to prevent cases from being diagnosed late?

These are two settings that need different approaches to prevention and control.

2)                  More data about ecological conditions of the main vectors incriminated in P. knowlesi transmission. Authors described the main vectors but information about vector capacity like larval habitats, biting behaviour, etc. is important for prevention and control measures. As several anopheline species can be P. knowlesi transmitters, it is necessary to identify whether there is a difference between the different species (potential larval habitats, biting behavior, etc.) Is there information? What's left to know?

Specific comments

·       Throughout the document, the initial of the genus must be corrected: Plasmodium, Anopheles with capital in the first letter;

·       Sentences of lines 61-62 and lines 52-53 are repeated;

·       Line 66: BFMP: The first time that initials are cited in a text, the full name must be added;

·       Line 113: What do the authors want to say when describe “parameter spraying”? It is not clear;

·       Line 182-183: The larval habitats of each species are poorly described; preventive measures will depend of the context. For example, what will be the benefit of modifying the habitats within the jungle. Is that possible? The suggestion is to context more each setting (inside the jungle, peri-domestic, inside the houses);

·       Line 218: Authors said that: “…the mosquitoes are found to go inside the house 218 at one point in their life”. Please add a reference about this topic;

·       Line 314: Authors “acknowledge the Faculty of Medicine and Health Sciences for support given”. What Institution? Where is it located? What country?

·       Line 318-512: Format the references. There are some that are cited incompletely. Put all scientific names in italics and capitalize the first letter of the genus.

Author Response

We appreciate the comments from the reviewer. 

We humbly believe it would help to improve the quality of this paper.

Revision and comments have been edited in the main text and tracked.

Reviewer 2 Report

Authors wrote an interesting paper deeply treated. I appreciate a lot and also for me was opportunity to appreciate it. 

Below my suggestions

1. Introduction: update data on Malaria burden with data from Malaria word report 2021. 

2. PREVENTION AND CONTROL OF KNOWLESI MALARIA: add data on therapeutic resistance and also on net and insecticide resistance 

Add also the role of SARS CoV2 pandemic on disruption of Malaria services and the negative impact on Malaria control (see Malaria and COVID-19: Common and Different Findings. Trop Med Infect Dis. 2020 Sep 6;5(3):141. doi: 10.3390/tropicalmed5030141. PMID: 32899935; PMCID: PMC7559940.)

3. Add methods section

4. Discuss better the role also of diagnostic test and the IPT in pregnancy

5. Conclusion give some global health proposal that came from your interesting literature review 

Author Response

Dear Sir,

We highly appreciate the review done for this article.

Therefore, we include here with the track changes for your kind perusal.

Reviewer 3 Report

It is interestng article and well formulated. There are missing figures which could perhaps enhance the readibility. I also suggest the author to discuss about the reasons emerging diagnostic ttechnologies

Nature medicine 20 (9), 1069

https://www.nature.com/articles/s41467-021-21110-w

Communications Biology 3 (1), 1-10

Diagnostics 11 (12), 2222

Engineering Reports 3 (10), e12383

Additional comments:

1. Can you comment why Plasmodium knowlesi instead of Plasmodium falciparum is discussed ?

2. Maybe explain how different or similar anti malarial drugs used for treatment.

3. Suggest using sub titles to make this essay readable. 

4. Line 212 Rural should be rural 5. Line 232 should be N, N-diethyl-m-toluamide (DEET)

Author Response

Dear reviewer, 

Thank you so much for the good comments.

We greatly appreciate it.

We attached herewith the edited version.

Thank you

Round 2

Reviewer 1 Report

It is a very interesting review about control and prevention of zoonotic malaria by P. knowlesi, a Plasmodium of monkeys of Southeast Asia. The second version of the manuscript, in general, is well written and essential questions were answered by the authors.

This new version requires a careful revision of English language.  I don´t have more comments. 

Author Response

We have proof read the manuscript based on the feedback from the reviewer.
